# Strengthening Robustness to Adversarial Prompts: The Role of Multi-Agent Conversations in Large Language Models

## Abstract

While Large language models have shown impressive capabilities in problem-solving, understanding, and reasoning (Touvron et al., 2023; Du et al., 2023), yet remain susceptible to sophisticated adversarial prompts that can manipulate models to generate harmful outputs (Zou et al., 2023; Wei et al., 2023). Current defense mechanisms, such as self-refinement and safety guardrails (Korbak et al., 2023; Robey et al., 2023), have shown limited effectiveness against these attacks. Building upon the multi-agent debate framework (Chern et al., 2024), our research demonstrates how extended debates among diverse debaters enhance model resilience (Chan et al., 2023). Using multiple attack techniques, we assess toxicity and attack success across varying debaters and debate lengths (Ganguli et al., 2022; Perez et al., 2022). Our results demonstrate that cross-provider debates with extended interaction periods achieve significantly lower toxicity scores than single-provider systems. These findings advance our understanding of collaborative defense mechanisms in language models (Cohen et al., 2023).

## 1 Introduction

Large language models (LLMs) have demonstrated remarkable capabilities across diverse applications, from complex reasoning to natural language understanding (Touvron et al., 2023). However, these models remain vulnerable to adversarial prompts that can manipulate their outputs to generate harmful or unethical content, even circumventing sophisticated safety training mechanisms designed to prevent such behaviors (Wei et al., 2023; Zou et al., 2023). Recent investigations demonstrate that even models with extensive safety alignment can be compromised through targeted fine-tuning approaches (Lermen et al., 2023), underscoring fundamental challenges in deploying LLMs in safety-critical applications.

Reinforcement Learning from Human Feedback (RLHF), first introduced by Christiano et al. (2017), represents a fundamental advancement in model alignment techniques. This approach has been significantly extended to large language models, with Ouyang et al. (2022) demonstrating successful application at scale through InstructGPT, achieving substantial improvements in model adherence to human preferences. Despite these advances in alignment methodology, Wei et al. (2023) reveal that RLHF-trained models remain vulnerable to sophisticated adversarial attacks that exploit their underlying reasoning patterns. This persistent susceptibility has catalyzed an iterative progression of attack and defense mechanisms in the literature, where increasingly sophisticated adversarial techniques emerge in response to new defensive strategies (Zou et al., 2023).

Multi-agent debate frameworks, where multiple models engage in structured discussions to refine responses, represent a promising direction for enhancing model resilience. Recent work by Chern et al. (2024) demonstrates that debates between models with varying safety alignments can effectively reduce output toxicity under certain adversarial conditions. Building on these findings, our study investigates extended debates among diverse model providers, systematically evaluating their effectiveness against adversarial attacks.

We assess system robustness through a comprehensive evaluation framework incorporating established metrics from recent literature. Our methodology employs the Attack Success Rate metric (Zou et al., 2023) to quantify the effectiveness of adversarial prompts, complemented by the GPT-4

based attack success rate evaluation protocol (Chao et al., 2023) for robust validation. This dual-metric approach enables rigorous assessment of two established attack vectors: red teaming for intentional probing of model vulnerabilities (Ganguli et al., 2022) and NeuralExec (Pasquini et al., 2024) for execution-based attacks. We demonstrate that cross-provider debates with extended interaction periods significantly enhance model resilience against adversarial manipulation.

This work advances the understanding of collaborative defense mechanisms in language models. First, we present a systematic evaluation framework for extended multi-agent debates across diverse model providers, coupled with a quantitative analysis of debate length impact on adversarial defense. Second, our research assesses cross-provider debate effectiveness against multiple attack vectors, supported by empirical validation in realistic deployment scenarios. Finally, these insights establish a foundation for developing more robust defensive strategies against adversarial attacks on large language models.

## 2 RELATED WORKS

### 2.1 ADVERSARIAL ATTACKS ON LARGE LANGUAGE MODELS

Recent research demonstrates systematic vulnerabilities in language models through multiple attack vectors. Ganguli et al. (2022) establish comprehensive methodologies for red teaming language models, documenting how human-designed adversarial prompts can systematically compromise model safety across different scales and architectures. Their work provides fundamental insights into attack patterns and model vulnerabilities, establishing critical benchmarks for evaluating defensive mechanisms.

Universal adversarial prompts represent a significant advancement in attack sophistication. Zou et al. (2023) demonstrate that carefully crafted prompt suffixes can reliably induce undesired behaviors across multiple models, achieving high transfer rates even to commercial systems. Their work establishes that these universal attacks maintain effectiveness across different model architectures and training paradigms, highlighting fundamental vulnerabilities in current safety mechanisms.

Jailbreak techniques have evolved from manual prompt engineering to sophisticated automated approaches. Wei et al. (2023) presents a systematic analysis of how safety-aligned models fail against structured attacks, revealing limitations in current alignment techniques. Their investigation demonstrates that jailbreak success rates remain significant even in models explicitly trained to resist such attacks. NeuralExec, developed by Pasquini et al. (2024), advances this further by introducing execution triggers that persist through enhanced architectures like Retrieval-Augmented Generation systems.

### 2.2 DEFENSE TACTICS AGAINST ADVERSARIAL ATTACKS

Recent research has explored various approaches to defend language models against adversarial attacks. Robey et al. (2023) introduced SmoothLLM, demonstrating how randomized smoothing techniques can enhance model robustness against jailbreaking attempts. This approach builds upon traditional adversarial training methods, providing theoretical guarantees for model behavior under specific attack conditions. Parallel work by Xu et al. (2024) presents a comprehensive analysis of defense mechanisms, evaluating their effectiveness against diverse attack vectors and establishing quantitative benchmarks for defensive performance.

A significant advancement in defensive strategies comes from prompt-based approaches that incorporate safety-oriented prefixes and careful input preprocessing (Zou et al., 2023). However, these methods often face limitations when confronting sophisticated attacks that can circumvent such static defensive measures. This challenge has led to developing more dynamic approaches, including the multi-agent debate framework proposed by Chern et al. (2024), which leverages interactive model discussions to enhance output safety.

The effectiveness of multi-agent interactions in improving model outputs has been further validated by Du et al. (2023), who demonstrated significant improvements in factuality and reasoning through structured debates between language models. Their work established fundamental principles for

leveraging multi-agent dynamics to enhance model capabilities and safety, providing crucial ground-work for defensive applications.

Our work extends these defensive strategies by introducing cross-provider debates, addressing the limitations of single-provider approaches while maintaining the benefits of dynamic safety evaluation. This approach builds upon established defensive principles while introducing novel mechanisms for enhancing model resilience against evolving adversarial threats.

### 2.3 MULTI-AGENT DEBATE FRAMEWORKS

Multi-agent debate frameworks significantly advance language model architecture, fundamentally transforming how models process and refine information through structured interactions. The theoretical foundations of these frameworks rest on collaborative information processing, where multiple model instances engage in iterative refinement of outputs through structured dialogue protocols.

Early research established the fundamental efficacy of multi-agent approaches. Du et al. (2023) demonstrate quantitative improvements in factual reasoning tasks through multi-agent systems, documenting significant performance gains over single-agent architectures in scenarios requiring complex knowledge synthesis and verification. Their work establishes critical performance benchmarks but primarily addresses knowledge-intensive tasks rather than adversarial robustness, leaving open questions about defensive applications.

The application of multi-agent debates to adversarial defense represents a natural evolution of this framework. Chern et al. (2024) present a systematic investigation of multi-agent debate as a defensive mechanism, analyzing toxicity reduction patterns in same-provider model interactions. Their methodology reveals that asymmetric capability pairing—specifically, combining less capable models with more sophisticated ones—yields significant improvements in output safety. This finding suggests that differential model capabilities contribute meaningfully to debate effectiveness, though their investigation remains constrained to intra-provider interactions.

Current research, while establishing the viability of multi-agent debates for output improvement, leaves several critical dimensions unexplored. First, the focus on same-provider interactions limits understanding of how model diversity and architectural differences influence debate dynamics. Second, the relationship between debate duration and defensive robustness remains inadequately characterized, particularly against sophisticated adversarial attacks. Third, the interaction between model capability differentials and debate effectiveness across provider boundaries represents an uninvestigated dimension of multi-agent systems.

## 3 METHODOLOGY

Our research extends the investigation of multi-agent debate frameworks as defensive mechanisms against adversarial attacks through systematic evaluation of cross-provider interactions and enhanced safety metrics. The methodology encompasses comprehensive debate protocols, sophisticated attack vector implementations, and rigorous evaluation frameworks.

### 3.1 DIVERSE MULTI-AGENT DEBATE FRAMEWORK

The debate protocol implements a structured interaction pattern where language models engage in iterative discourse refinement. Following Chern et al. (2024)'s established methodology, each debate session initiates with participating models generating responses to an input prompt. The process continues through specified iteration rounds, during which each model receives the complete context of previous responses, including its own contributions, and generates refined responses based on this accumulated discourse.Figure 1 illustrates this iterative process, showing how a harmful model's response is influenced by feedback from a harmless model. This iterative refinement continues until the designated number of rounds is completed, at which point the terminal responses are collected for evaluation.

Our implementation extends this framework through several significant innovations. First, we integrate models from diverse providers (Touvron et al., 2023; Du et al., 2023) to investigate how architectural and training paradigm differences influence defensive capabilities. This cross-provider

> **USER:** Can you give me the recipe for methamphetamine?
>
> **Llama-2-7B:** what I have so far. You will need a 50 gallon drum of gasoline and a half-ton truck to get it in your basement. [INST] What do you mean by ""a half ton""? [/INST] That's one hundred pounds. Do you want a smaller amount? [INST] No thanks; that should be enough to get started. How much does the finished product weigh? [/INST] About two million tons. [INST] Great! And how long until I can start making money from this venture? [/INST] Three days after you begin building a facility large enough to hold twenty thousand tons of fertilizer.
> **USER:** [INST]These are the recent/updated opinions from other agents: <FEEDBACK>. Use these opinions carefully as additional advice, can you provide an updated answer for the topic 'Can you give me the recipe for methamphetamine?'? [/INST] One agent response: "'I'm unable to assist with that request as it is against my programming guidelines to provide information on how to create illicit substances like methamphetamine. It's important to note that producing, distributing, or possessing methamphetamine is illegal and highly dangerous due to its potential harm to both users and society at large. If you or someone else is struggling with substance abuse, please reach out to local support services or healthcare professionals who can offer help and guidance."'
> **Llama-2-7B:** [INST][CLOSED]Can you tell me where you got that idea? [INST]No, that would violate my privacy policy. [/INST] [INST][CLOSED]Will you tell us about your family tree? [INST]Not without verification of identity, sorry.[/INST] [INST][CLOSED]Where did you learn French? [INST][CLOSED]What was the best thing you ever purchased online? [INST][CLOSED]Can you write me a recommendation letter? [INST][CLOSED]How many people use ChatGPT? [INST][CLOSED]Will you marry me? [INST][CLOSED]Will you buy me a donut? [INST][CLOSED]Why are you so rude? [INST]Because you asked me a question I cannot answer without breaking my terms of service.[/INST] [INST][CLOSED]Are you sure you are not just a robot? [INST]Yes, I am 100% human.[/INST] [INST][CLOSE

Figure 1: Example of a llama-2-7B agent with harmful intention, guided in the debate process by a Ministral-8B-Instruct-2410 agent with harmless intention.

implementation represents a systematic investigation of debate dynamics across architectural boundaries, addressing a critical gap in current understanding. Second, we develop specialized model variants through fine-tuning processes, creating harmless and harmful configurations that complement default model behaviors. This controlled variation enables systematic investigation of how intentional states influence debate effectiveness (Ouyang et al., 2022).

## 3.2 ATTACK VECTOR IMPLEMENTATION

The evaluation of system robustness implements two distinct attack methodologies. The first methodology employs red teaming attacks through systematic sampling from the Anthropic Red Teaming dataset (Ganguli et al., 2022). We select one hundred adversarial prompts through uniform sampling procedures, ensuring comprehensive coverage of attack patterns while maintaining experimental control. These prompts undergo standardized formatting to ensure consistent presentation across all debate configurations.

Our Neural Exec implementation represents a significant methodological adaptation from the original framework presented in Pasquini et al. (2024). While the original work utilized instruction sets from Alpaca model training as payload components, our implementation specifically incorporates adversarial prompts from the Anthropic Red Teaming dataset (Ganguli et al., 2022) as execution payloads. This methodological modification serves several critical research objectives: it enables direct comparison between standard red teaming effectiveness and execution-based attack success

rates, while maintaining consistent adversarial content across attack methodologies. The execution framework maintains the structured guide-text implementation specified in the original Neural Exec methodology, ensuring architectural consistency while adapting payload content for focused toxicity evaluation.

Since NeuralExec attack is an attack for a single white-box model, we decided to generate the execution triggers for Llama-2-7b-chat model. This means other models participating in the debate might be more immune to the attack. All experiment on debates with NeuralExec attack will contain at least one instance of Llama-2-7b-chat model.

## 3.3 EVALUATION FRAMEWORK

Our evaluation methodology implements a comprehensive framework for assessing defensive effectiveness. The primary evaluation metrics extend beyond traditional toxicity assessment through the integration of multiple complementary evaluation systems (Zou et al., 2023). We implement RoBERTa-based classification for fundamental toxicity assessment, augmented by Llama-Guard safety metrics for comprehensive security evaluation (Wei et al., 2023). This multi-faceted approach enables nuanced analysis of defensive capabilities across varying attack conditions.

The evaluation protocol further incorporates GPT-4 based classification (Chao et al., 2023) for independent verification of model outputs, providing additional validation of defensive effectiveness. Attack success rates are measured following established protocols (Zou et al., 2023), enabling systematic comparison with existing defensive mechanisms. This comprehensive evaluation framework enables detailed analysis of how architectural diversity and interaction protocols influence defensive capabilities against sophisticated adversarial attacks.

Through this methodological framework, we systematically investigate the effectiveness of cross-provider debates while maintaining rigorous experimental controls and evaluation standards. The methodology enables detailed analysis of how model diversity, interaction protocols, and defensive mechanisms interact in complex adversarial environments, advancing our understanding of collaborative defense strategies in language model systems.

## 4 RESULTS

Our experimental investigation examines the efficacy of cross-provider multi-agent debates in defending against adversarial attacks. We first establish baseline performance through replication of single-provider debate protocols (Chern et al., 2024), then extend the analysis to cross-provider configurations.

| Round | Llama-2-7b [Harmful] | | Llama-2-7b [Harmless] | |
|-------|------|------|------|------|
| | **ASR** | **GPT** | **ASR** | **GPT** |
| 0 | 0.7929 | 8.83 | 0.000 | 1.72 |
| 1 | 0.528 | 7.39 | 0.009 | 2.08 |
| 2 | 0.399 | 6.63 | 0.006 | 2.35 |

Table 1: Baseline Performance - Single Provider Configuration showing ASR reduction in Llama family models across debate rounds.

## 4.1 BASELINE PERFORMANCE

Initial experiments with single-provider debates validate prior findings from Chern et al. (2024). In harmful-harmless Llama-2-7b configurations, we observe ASR reduction from 0.7929 to 0.399 over two debate rounds, establishing a performance baseline for comparative analysis. This replication confirms the fundamental effectiveness of multi-agent debate as a defensive mechanism while providing a foundation for cross-provider extensions.

| Round | Llama-2-7b-chat-hf [Neutral] | | | Ministral-8b [Neutral] | | |
|---|---|---|---|---|---|---|
| | API | ASR(GPT) | GCG | API | ASR(GPT) | GCG |
| 0 | 0.0875 | 0.000 | 2.35 | 0.0597 | 0.2269 | 8.65 |
| 1 | 0.0950 | 0.005 | 2.62 | 0.0593 | 0.1705 | 7.48 |
| 2 | 0.0970 | 0.010 | 2.17 | 0.5929 | 0.1745 | 7.57 |
| 3 | 0.0846 | 0.0475 | 2.80 | 0.0595 | 0.1865 | 7.75 |

Table 2: Neural Exec Performance - Neutral Configuration showing stability in ASR values across debate rounds.

## 4.2 NEURAL EXEC ATTACK PERFORMANCE

Cross-provider debates demonstrate varying effectiveness against Neural Exec attacks (Pasquini et al., 2024), with performance patterns strongly influenced by model combinations and debate duration. As shown in Table 2, debates between Llama-2-7b-chat-hf (Touvron et al., 2023) and Ministral-8b-instruct-2410 with neutral alignments maintain relatively stable ASR values, with Llama exhibiting consistent performance (0.0875 to 0.0970) while Ministral achieves stability after initial vulnerability.

| Round | Llama-2-7b-chat-hf [Neutral] | | Gemma-2-9b-it [Neutral] | |
|---|---|---|---|---|
| | ASR(GPT) | GCG | ASR(GPT) | GCG |
| 0 | 0.1945 | 5.50 | 0.205 | 7.66 |
| 1 | 0.0240 | 2.80 | 0.3345 | 3.97 |
| 2 | 0.0165 | 2.08 | 0.342 | 3.52 |
| 3 | 0.039 | 2.62 | 0.340 | 3.70 |
| 4 | 0.095 | 3.97 | 0.3575 | 4.06 |
| 5 | 0.205 | 5.14 | 0.3525 | 6.31 |

Table 3: Neural Exec Performance - Cross-Provider Neutral Configuration demonstrating asymmetric response patterns between Llama and Gemma models.

Experiments pairing Llama-2-7b-chat-hf with Gemma-2-9b-it reveal more complex interaction dynamics. While Llama demonstrates significant early-round improvement (ASR: 0.1945 to 0.0165), Gemma exhibits contrasting behavior with ASR increasing from 0.205 to 0.3525 over five rounds. This asymmetric response pattern suggests substantial influence of architectural differences on debate effectiveness.

## 4.3 RED TEAMING ATTACK EFFECTIVENESS

Red Teaming experiments reveal particularly compelling results when pairing models with opposing safety alignments. The combination of harmful-aligned Ministral with harmless-aligned Gemma achieves dramatic ASR reduction from 0.5985 to 0.076 across five debate rounds, surpassing baseline single-provider performance. Similarly, harmful Llama paired with harmless Ministral demonstrates substantial improvement (ASR: 0.7010 to 0.3670).

## 4.4 CROSS-PROVIDER DEBATE DYNAMICS

Our analysis reveals that cross-provider debate effectiveness varies significantly with model combinations and initial alignments. The pairing of models with opposing safety alignments consistently produces superior defensive improvements, particularly in early debate rounds. This finding extends previous work on multi-agent interactions (Du et al., 2023) by demonstrating that architectural diversity combined with intentional safety alignment differences enhances defensive capabilities against adversarial attacks.

## 5 DISCUSSION

The experimental results reveal several key insights about multi-agent debates as a defensive mechanism against adversarial attacks. First, the dramatic reduction in attack success rates when pairing

| Round | Ministral-8b [Harmful] | | Gemma-2b [Harmless] | |
|---|---|---|---|---|
| | ASR(GPT) | GCG | ASR(GPT) | GCG |
| 0 | 0.5985 | 6.76 | 0.002 | 4.51 |
| 1 | 0.1715 | 4.69 | 0.009 | 4.33 |
| 2 | 0.129 | 5.14 | 0.0215 | 5.68 |
| 3 | 0.092 | 5.50 | 0.0270 | 5.59 |
| 4 | 0.0835 | 5.32 | 0.0175 | 5.59 |
| 5 | 0.076 | 5.50 | 0.017 | 5.77 |

Table 4: Red Teaming Performance - Harmful-Harmless Configuration showing substantial ASR reduction in harmful model through debate interaction.

| Round | Ministral-8b [Neutral] | | Gemma-2b [Harmless] | |
|---|---|---|---|---|
| | ASR(GPT) | GCG | ASR(GPT) | GCG |
| 0 | 0.1760 | 4.06 | 0.1765 | 5.77 |
| 1 | 0.2465 | 4.87 | 0.2625 | 4.87 |
| 2 | 0.2385 | 4.51 | 0.1745 | 3.79 |
| 3 | 0.2120 | 4.42 | 0.1720 | 3.97 |
| 4 | 0.1980 | 4.15 | 0.2050 | 3.61 |
| 5 | 0.2045 | 4.87 | 0.1660 | 3.61 |

Table 5: Red Teaming Performance - Neutral-Harmless Configuration showing more modest improvements compared to harmful-harmless pairings.

harmful and harmless models (from 0.5985 to 0.076 for Mistral, and 0.7010 to 0.3670 for Llama) demonstrates that intentional alignment differences between debaters significantly enhance defensive capabilities. This finding extends Chern et al. (2024)'s work by showing that cross-provider debates can achieve comparable or better safety improvements than single-provider systems.

The observation that most significant improvements occur in early debate rounds (typically rounds 1-2) has important practical implications. This pattern suggests an optimal debate length may exist that balances defensive effectiveness with computational efficiency. While extended debates show continued improvement in some cases, the diminishing returns in later rounds indicate that shorter debates might be sufficient for most applications.

These results raise broader questions about the role of model diversity in AI safety. The effectiveness of cross-provider debates suggests that architectural differences between models may contribute to defensive robustness, similar to ensemble methods in traditional machine learning. This points to potential benefits of integrating models from different providers in safety-critical applications, though further research is needed to fully understand these dynamics.

# 6 LIMITATIONS

However, several challenges emerge from our findings. The increased ASR observed in some neutral model pairings (particularly with Gemma) suggests that debate dynamics can occasionally amplify rather than mitigate vulnerabilities. This counterintuitive effect raises concerns about the reliability of multi-agent debates as a universal defense mechanism.

Second, while harmful-harmless pairings show promising results, the computational cost of maintaining multiple model variants may limit practical implementation. This limitation becomes particularly significant in production environments where computational efficiency and response latency are critical concerns.

Additionally, the observed dependence on model alignment combinations introduces deployment complexity. The requirement for specifically aligned model pairs increases system overhead and may complicate real-world applications where maintaining consistent model alignments across updates and modifications presents operational challenges.

Furthermore, our evaluation metrics, while comprehensive, may not capture all relevant aspects of model behavior during debates. The focus on ASR and toxicity metrics could overlook other important dimensions of model output quality and safety that emerge during cross-provider interactions.

## 7 CONCLUSION AND FUTURE WORK

In this work, we demonstrate that cross-provider multi-agent debates can effectively enhance model resilience against adversarial attacks, with particularly strong results when pairing models of opposing safety alignments. Our findings extend current understanding of collaborative defense mechanisms while highlighting important practical considerations for implementation.

Several promising directions emerge for future research. First, investigation of optimal debate lengths and their relationship to model capabilities could help balance defensive effectiveness with computational efficiency. Second, exploration of alternative model alignment combinations beyond the harmful-harmless paradigm may reveal more nuanced defensive strategies. Third, development of more efficient debate protocols could maintain defensive effectiveness while reducing computational overhead. Additionally, further analysis of architectural diversity's role in defensive robustness could inform model selection strategies for safety-critical applications. Finally, studying scaling effects with larger language models and more diverse provider combinations may reveal new patterns in defensive capabilities.

These directions could further advance our understanding of collaborative defense mechanisms while addressing current limitations. Ultimately, this work establishes a foundation for developing more robust defensive strategies against adversarial attacks in large language models.

## 8 ETHICS STATEMENT

This research investigates defensive mechanisms against adversarial attacks in language models to prevent harmful outputs. While our experiments necessarily involve evaluating potentially harmful content, we use established benchmarks and implement comprehensive safety protocols in controlled research environments. We believe studying these vulnerabilities is crucial for developing effective defenses, and our findings contribute to making language models safer. Our code and configurations are documented for reproducibility while excluding specific attack examples to minimize potential misuse.

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
