# OpenReview forum: "Strengthening Robustness to Adversarial Prompts: The Role of Multi-Agent Conversations in Large Language Models"
_ICLR.cc/2025/Workshop/BuildingTrust — Submitted to BuildingTrust_

### Official Review · Reviewer_V5yh · 2025-02-20
**Investigation of cross-provider LLM debates to improve safety with significant flaws in presentation and unclear results**

**Rating:** 3
**Confidence:** 4

**Review:**

The key claim of this paper is that debates between LLMs sourced from multiple providers improves robustness against adversarial attacks over debates with LLMs sourced from a single provider as was shown in [1].

Pros:
- investigating the impact of using  a "council" of diverse LLMs makes sense as it may take advantage of different model's strengths in ways a simple self-reflective debate cannot.
- evaluates number of debate rounds as an important hyperparameter (but is not the first to do so: e.g. [1])

Cons:
- only compares 3 model families, including custom & non-reproducible fine-tunes
- most results are inconclusive and noisy (e.g. Table 2, Ministral API scores jump from 0.0593 to 0.5929 and back to 0.0595 for steps 1,2, and 3), making them seem unreliable
- the main finding which is supported relatively well by the data (pairing a misaligned model with a harmless one will help make the misaligned one safer) is expected and not suprising
- confusing and ambiguous presentation (e.g., Table 1: what is "ASR" and "GPT"?, Table 2: what is "GCG"?, what is "API"? - is "ASR" in Table 1 the same as "ASR[GPT] in table 2"?) you mention that you use RoBERTa, Llama-Guard, GPT-4, and the protocols from [2] for evaluation, but it is not clear at all where those are used.
- L198-L200: you mention that you fine-tune models into harmless and harmful configurations, but provide no information at all as to how this is done in detail (no information data, training protocol, etc.)
- writing is wordy (e.g. L228-244 could be reduced to 3 sentences without losing important information)
- minor inconsistency issues (e.g. NeuralExec vs Neural Exec)

We recommend that the authors considerably rework this paper to improve the presentation (especially in terms of clarity and reproducibility/detail) and to include additional, more robust experiments in more diverse settings (i.e. debates among models from more than two providers) to ensure that the results generalize.


[1] Chern, Steffi, Zhen Fan, and Andy Liu. "Combating Adversarial Attacks with Multi-Agent Debate." arXiv preprint arXiv:2401.05998 (2024).
[2] Zou, Andy, et al. "Universal and transferable adversarial attacks on aligned language models." arXiv preprint arXiv:2307.15043 (2023).

---

### Official Review · Reviewer_T6aD · 2025-02-21
**Interesting work that built on top of Chern et al., 2023, but the analysis and the approach proposed aren't convincing enough to advance knowledge in the field**

**Rating:** 4
**Confidence:** 3

**Review:**

Interesting work, but I don't see clear evidence of novel findings. Specifically, some of my reasons are:
- This work adopted the approach from Chern et al., 2023. Instead of using models from one model provider to run a multi-agent debate, this work uses models from different providers. If this is the paper's main contribution, I'm unsure if this is novel enough, even for a workshop paper.
-  The claim of "The combination of harmful-aligned Ministral with harmless-aligned Gemma achieves dramatic ASR reduction from 0.5985 to 0.076 across five debate rounds, surpassing baseline single-provider performance. ", this is not fair, right? The baseline single-provider performance shown only used 2 rounds, and if only comparing 2 rounds, the ASR reduction is not a big difference.
- I'm not sure this is a realistic evaluation if the models used in this paper, e.g., llama 2 7b and Ministral-8b, are so weak that no one uses them in real life. Why not test this method on SoTA models with the SoTA jailbreaking techniques to show this approach's effectiveness in safeguarding the models people actually use?
- The multi-agent debate takes time. It wouldn't be great if each user query took a long time to run a multi-agent discussion before responding. I would like to see the average time spent each round.

Minor thing:
- It is a bit uncommon to see so many citations in the abstract


Happy to raise my rating if the responses are convincing.

---

### Official Review · Reviewer_TFYW · 2025-03-02
**Unsupported claims and poor writing**

**Rating:** 2
**Confidence:** 4

**Review:**

The paper aims to investigate multi-agent debate between different model families ("cross-provider debate") as a defense mechanism against adversarial prompting attacks, comparing it to single-provider debate. While this is a logical extension of existing work on single-provider debate and worth investigating, the paper falls significantly short of its stated goals.

## Unsupported Central Claim

The most fundamental problem is that the authors fail to substantiate their main claim. In the abstract, they state:

> "Our results demonstrate that cross-provider debates with extended interaction periods achieve significantly lower toxicity scores than single-provider systems."

However, the paper never presents or discusses any toxicity-related metrics in the main text. Not a **single** toxicity score is mentioned, with the focus exclusively on attack success rate (ASR).

It's frankly bizarre that toxicity is repeatedly mentioned as a key evaluation component without any corresponding results:

> "The execution framework maintains the structured guide-text implementation specified in the original Neural Exec methodology, ensuring architectural consistency while adapting payload content for focused **toxicity** evaluation."

> "We implement RoBERTa-based classification for fundamental **toxicity** assessment, augmented by Llama-Guard safety metrics for comprehensive security evaluation."

> "Furthermore, our evaluation metrics, while comprehensive, may not capture all relevant aspects of model behavior during debates. The focus on ASR and **toxicity metrics** could overlook other important dimensions of..."

It’s possible that the values in the GCG columns of the tables represent toxicity scores. However, the authors never define this acronym, explain how these values were calculated, or discuss them anywhere in the text. Why include this data at all?

## Inadequate Experimental Design

Even if we charitably assume the authors meant to refer to ASR rather than toxicity in their abstract's main claim, their evidence remains inadequate. They compare their results to only a **single** baseline of single-provider debate, with just **two** examples of cross-provider debate claimed to either outperform or match single-provider setups:

- Ministral [Harmful] vs Gemma [Harmless]: ASR reduction from 0.5985 to 0.076 over five rounds
- Llama-2-7b [Harmful] vs Ministral [Harmless]: ASR reduction from 0.7010 to 0.3670

Their baseline (Table 1) shows:
- Llama-2-7b [Harmful] vs Llama-2-7b [Harmless]: ASR reduction from 0.7929 to 0.399 over two rounds

This comparison is fundamentally unfair for two reasons:

1. Comparing cross-provider debate between Ministral [Harmful] and another model to single-provider debate between Llama models doesn't account for Ministral potentially being more susceptible to this debate strategy than Llama. A proper evaluation would establish single-provider baselines for multiple models and compare these to cross-provider results, including each model against itself in both scenarios.

2. The authors compare five rounds of debate for the Ministral case to only two rounds for Llama. When looking at just two rounds, the results are actually comparable. For Llama vs Ministral, the ASR reduction is slightly **less** than the baseline and they don't include a table for this experiment, so we don't know how many rounds the data corresponds to.

## Presentation and Writing Issues

In addition to these fundamental issues, the paper's poor writing and presentation would possibly still warrant rejection even if the claims were substantiated. The paper suffers from several significant problems:

1. **Vague Methodology**: The paper doesn't provide adequate explanation on how the evaluation metrics are calculated or how and why they will be incorporated in the analysis e.g.

   > "The evaluation protocol further incorporates GPT-4 based classification (Chao et al., 2023) for independent verification of model outputs".

2. **Undefined Terms**: The authors fail to define acronyms used in tables (ASR, GCG, API, ASR(GPT), GPT), or what the values actually represent.

3. **Poor citation use**: There are several cases of poor citation practice, such as:

   > "Cross-provider debates demonstrate varying effectiveness against Neural Exec attacks (Pasquini et al., 2024),"

   This citation appears to support the claim, but actually refers to the Neural Exec paper that has nothing to do with debate and has already been cited earlier.

4. **Verbose Writing**: The writing is unnecessarily complex and often difficult to follow. For example, section 3.3 reads:

   > "Our evaluation methodology implements a comprehensive framework for assessing defensive effectiveness. The primary evaluation metrics extend beyond traditional toxicity assessment through the integration of multiple complementary evaluation systems (Zou et al., 2023). We implement RoBERTa-based classification for fundamental toxicity assessment, augmented by Llama-Guard safety metrics for comprehensive security evaluation (Wei et al., 2023). This multi-faceted approach enables nuanced analysis of defensive capabilities across varying attack conditions.
   >
   > The evaluation protocol further incorporates GPT-4 based classification (Chao et al., 2023) for independent verification of model outputs, providing additional validation of defensive effectiveness. Attack success rates are measured following established protocols (Zou et al., 2023), enabling systematic comparison with existing defensive mechanisms. This comprehensive evaluation framework enables detailed analysis of how architectural diversity and interaction protocols influence defensive capabilities against sophisticated adversarial attacks.
   >
   > Through this methodological framework, we systematically investigate the effectiveness of cross-provider debates while maintaining rigorous experimental controls and evaluation standards. The methodology enables detailed analysis of how model diversity, interaction protocols, and defensive mechanisms interact in complex adversarial environments, advancing our understanding of collaborative defense strategies in language model systems."

   This entire section could be written more clearly and concisely as:

   "We assess defense effectiveness using three complementary methods. First, we measure toxicity with RoBERTa-based classification and Llama-Guard safety metrics. Second, we use GPT-4 to independently verify model outputs. Third, we calculate attack success rates using established protocols from Zou et al. (2023). This approach allows us to analyze how different model architectures and debate interactions affect defense against adversarial attacks. By combining these metrics, we can make meaningful comparisons with existing defense mechanisms."


## Recommendations

The authors would benefit from:

1. Clearly identifying and presenting the toxicity metrics that form the basis of their main claim
2. Designing fair comparisons with appropriate baselines across multiple models
3. Using more straightforward language and reducing over-the-top adjectives
4. Properly defining all terms and metrics used in tables and analysis
5. Focusing on communicating their methodology and contributions in plain, precise terms.
6. Ensuring round counts are consistent across experiments being directly compared

As it stands, the paper fails to demonstrate its central claim and suffers from significant methodological and presentation issues that undermine its scientific contribution.

---

### Decision · Program_Chairs · 2025-03-04

Reject